# The presence and absence of periplasmic rings in bacterial flagellar motors correlates with stator type

Mohammed Kaplan[1], Debnath Ghosal[1], Poorna Subramanian[1], Catherine M Oikonomou[1], Andreas Kjaer[1†], Sahand Pirbadian[2], Davi R Ortega[1], Ariane Briegel[1‡], Mohamed Y El-Naggar[2], Grant J Jensen[1,3*]

[1]Division of Biology and Biological Engineering, California Institute of Technology, Pasadena, United States; [2]Department of Physics and Astronomy, Biological Sciences, and Chemistry, University of Southern California, Los Angeles, United States; [3]Howard Hughes Medical Institute, California Institute of Technology, Pasadena, United States

**\*For correspondence:**
jensen@caltech.edu

**Present address:** [†]University of Oxford, Oxford, United Kingdom; [‡]Institute of Biology, Leiden University, Leiden, The Netherlands

**Competing interests:** The authors declare that no competing interests exist.

**Abstract** The bacterial flagellar motor, a cell-envelope-embedded macromolecular machine that functions as a cellular propeller, exhibits significant structural variability between species. Different torque-generating stator modules allow motors to operate in different pH, salt or viscosity levels. How such diversity evolved is unknown. Here, we use electron cryo-tomography to determine the in situ macromolecular structures of three Gammaproteobacteria motors: *Legionella pneumophila*, *Pseudomonas aeruginosa*, and *Shewanella oneidensis*, providing the first views of intact motors with dual stator systems. Complementing our imaging with bioinformatics analysis, we find a correlation between the motor's stator system and its structural elaboration. Motors with a single $H^+$-driven stator have only the core periplasmic P- and L-rings; those with dual $H^+$-driven stators have an elaborated P-ring; and motors with $Na^+$ or $Na^+/H^+$-driven stators have both their P- and L-rings embellished. Our results suggest an evolution of structural elaboration that may have enabled pathogenic bacteria to colonize higher-viscosity environments in animal hosts.
DOI: https://doi.org/10.7554/eLife.43487.001

## Introduction

The bacterial flagellum is a macromolecular machine that transforms the movement of ions ($H^+$, $Na^+$ or other cations) across the cell membrane into a mechanical torque to move the bacterial cell through its environment (*Ito and Takahashi, 2017*; *Sowa and Berry, 2008*). In general, the flagellum consists of a cell-envelope-embedded motor, a hook which acts as a universal joint and a long propeller-like filament (*Berg, 2003*; *Erhardt et al., 2010*). The motor is composed of a rotor and a stator: while the stator is the part of the motor that remains static, the rotor is the part that rotates and can rotate the filament in either a counterclockwise or clockwise direction. For cells with a single flagellum this drives the cell forward or backward; for peritrichous cells this results in 'run' or 'tumble' movements. Flagella can also exhibit a more complex behavior; it was recently reported that the *Shewanella putrefaciens* flagellum can wrap around the cell to mediate a screw-like motion that allows the cell to escape narrow traps (*Kühn et al., 2017*). Besides their role in motility, bacterial flagella participate in other vital activities of the cell such as biofilm formation (*Belas, 2014*). Moreover, the virulence of many human pathogens depends directly on their flagella, with flagellated strains of *Pseudomonas aeruginosa* and *Legionella pneumophila* causing more serious infections with higher mortality rates (*Appelt and Heuner, 2017*; *Feldman et al., 1998*). *P. aeruginosa* lacking fully-

**eLife digest** Bacteria are so small that for them, making their way through water is like swimming in roofing tar for us. In response, these organisms have evolved a molecular machine that helps them move in their environment. Named the bacterial flagellum, this complex assemblage of molecules is formed of three main parts: a motor that spans the inner and outer membranes of the cell, and then a 'hook' that connects to a long filament which extends outside the bacterium. More precisely, the motor is formed of the stator, an ion pump that stays still, and of a rotor that can spin. Different rings can also be present in the space between the inner and outer membranes (the periplasm) and surround these components. The stator uses ions to generate the energy that makes the rotor whirl. In turn, this movement sets the filament in motion, propelling the bacterium. Depending on where the bacteria live, the stator can use different types of ions. In addition, while many species have a single stator system per motor, some may have several stator systems for one motor: this may help the microorganisms move in different conditions.

As microbes colonize environments with a different pH or viscosity, they constantly evolve new versions of the motor which are more suitable to their new surroundings. However, a part of the motor remains the same across species. Overall, it is still unclear how bacterial flagella evolve, but examining the structure of new motors can shed light upon this process. Here, Kaplan et al. combine a bioinformatics approach with an imaging technique known as electron cryo-tomography to dissect the structure of the flagellar motor of three species of bacteria with different stator systems, and compare these to known motors of the same class.

The results reveal a correlation between the nature of the stator system and the presence of certain elements. Stators that use sodium ions, or both sodium and hydrogen ions, are associated with two periplasmic rings surrounding the conserved motor structure. These rings do not exist in motors with single hydrogen-driven stators. Motors with dual hydrogen-driven stators are, to some extent, an 'intermediate state', with only one of those rings present. As all the studied species currently exist, it is difficult to know which version of the motor is the most ancient, and which one has evolved more recently.

Capturing the diversity of bacterial motors gives us insight into the evolutionary forces that shape complex molecular structures, which is essential to understand how life evolved on Earth. More practically, this knowledge may also help us design better nanomachines to power microscopic robots.

DOI: https://doi.org/10.7554/eLife.43487.002

assembled flagella cause no mortality and are 75% less likely to cause pneumonia in mice (*Feldman et al., 1998*).

The best-studied flagellar motor, in *Salmonella enterica*, consists of several sub-complexes, which we will describe in order from the inside out. On the cytoplasmic side are the inner-membrane-embedded MS ring (formed by the protein FliF) and the C-ring (aka the switch complex, formed by FliN, FliM and FliG). The C-ring encircles a type III secretion system (T3SS) export apparatus (FliH, FliI, FliJ, FlhA, FlhB, FliP, FliQ and FliR). Spanning the space from the inner membrane to the peptidoglycan cell wall is the ion channel (called the stator), a complex of two proteins (MotA and MotB) with 4:2 stoichiometry (*Koebnik, 1995*; *Kojima, 2015*; *Morimoto and Minamino, 2014*). This complex is anchored to the peptidoglycan and converts the flux of ions across the bacterial membrane into a torque through the interaction of the so-called 'torque-helix' in MotA with FliG in the C-ring. Previous studies have shown that cycles of protonation/deprotonation of a certain aspartate residue in the cytoplasmic end of MotB induce conformational changes in MotA, which in turn interacts with the C-terminus of FliG. How the torque that is generated through this interaction is transferred to the other parts of the motor remains unclear with different models suggested (see *Berg, 2003*; *Stock et al., 2012* and references therein for details). The MS ring is coupled to the extracellular hook (FlgE) through the rod (FlgB, FlgC, FlgF and FlgG). The rod is further surrounded by two other rings: the P- (peptidoglycan, FlgI) and the L- (lipopolysaccharide, FlgH) rings which act as bushings during rod rotation. Extending from the hook is the filament (FliC) which is many micrometers in length. In addition to these components, the assembly of the whole flagellar motor is a highly

synchronized process that requires a plethora of additional chaperones and capping proteins (*Altegoer and Bange, 2015*; *Evans et al., 2014*; *Jones and Macnab, 1990*; *Kaplan et al., 2018*; *Kubori et al., 1992*; *Macnab, 1999*).

Recently, the development of electron cryo-tomography (ECT) (*Gan and Jensen, 2012*; *Oikonomou and Jensen, 2017*; *Pfeffer and Förster, 2018*) has allowed the determination of the complete structures of flagellar motors in their cellular milieu at macromolecular (~5 nm) resolution. ECT studies of many different bacterial species have revealed that while the core structure described above is conserved, the flagellar motor has evolved many species-specific adaptations to different environmental conditions (*Beeby et al., 2016*; *Chaban et al., 2018*; *Chen et al., 2011*; *Minamino and Imada, 2015*; *Terashima et al., 2017*; *Zhao et al., 2014*; *Zhu et al., 2017*). For example, extra periplasmic rings were found to elaborate the canonical P- and L-rings in the motor of the Gammaproteobacteria *Vibrio* species. These rings are called the T-ring (MotX and Y) and H-ring (FlgO, P and T) (*Terashima et al., 2010*; *Terashima et al., 2006*). Unlike the *S. enterica* motor described above, which is driven by $H^+$ ions, the motors of *Vibrio* and other marine bacteria employ different stators (PomA and PomB) which utilize $Na^+$. These $Na^+$-dependent stators generate higher torque (~2200 pN) than $H^+$-dependent stators (~1200 pN), driving the motor at higher speeds (up to 1,700 Hz compared to ~300 Hz in $H^+$-driven motors) (*Magariyama et al., 1994*).

Many flagellated bacteria use a single stator system – either $H^+$-driven or $Na^+$-driven, depending on their environment. Interestingly, it has also been shown that the single stator system of *Bacillus clausii* KSM-K16 is able to use both $Na^+$ and $H^+$ at different pH levels (*Terahara et al., 2008*). Additionally, some species (like alkaliphilic *Bacillus alcalophilus* AV1934 and *Paenibacillus* sp. TCA20), can use other cations to generate the energy required for torque generation depending on their environment (*Ito and Takahashi, 2017*; *Terahara et al., 2012*). Some species, however, such as *Vibrio alginolyticus*, use two distinct types of motors to move in different environments: a polar $Na^+$-driven flagellum and lateral $H^+$-driven flagella. Still other species employ dual stator systems with a single flagellar motor, conferring an advantage for bacteria that experience a range of environments (see *Thormann and Paulick, 2010* and references therein). For example, *P. aeruginosa* employs a dual $H^+$-driven stator system (MotAB and MotCD). While the MotAB system is sufficient to move the cell in a liquid environment (*Doyle et al., 2004*), MotCD is necessary to allow the cell to move in more viscous conditions (*Toutain et al., 2007*). *Shewanella oneidensis* MR-1 combines both $Na^+$- and $H^+$-dependent stators in a single motor, enabling the bacterium to move efficiently under conditions of different pH and $Na^+$ concentration (*Paulick et al., 2015*). How these more elaborate motors may have evolved remains an open question.

Here, we used ECT to determine the first in situ structures of three Gammaproteobacteria flagellar motors with dual stator systems: in *L. pneumophila*, *P. aeruginosa* and *S. oneidensis* MR-1. *L. pneumophila* and *P. aeruginosa* have dual $H^+$-dependent stator systems and *S. oneidensis* has a dual $Na^+$-$H^+$-dependent stator. This imaging, along with bioinformatics analysis, shows a correlation between the structural elaboration of the motor and its stator system, suggesting a possible evolutionary pathway.

## Results

To determine the structures of the flagellar motors of *L. pneumophila*, *P. aeruginosa,* and *S. oneidensis* we imaged intact cells of each species in a hydrated frozen state using ECT. We identified clearly visible flagellar motors in the tomographic reconstructions and performed sub-tomogram averaging to enhance the signal-to-noise ratio, generating a 3D average of the motor of each species at macromolecular resolution (*Figure 1* and *Figure 1—figure supplement 1*). Although the three motors shared the conserved core structure of the flagellar motor, they exhibited different periplasmic decorations surrounding this conserved core. While the *S. oneidensis* and *P. aeruginosa* averages showed clear densities corresponding to the stators (*Figure 1E,F,K and L*, dark orange density), none were visible in the *L. pneumophila* average, suggesting that they were more variable, or dynamic and therefore are not visible in the average (see e.g., (*Chen et al., 2011*; *Zhu et al., 2017*)). Interestingly, we observed a novel feature in the *S. oneidensis* motor: an extra ring outside the outer membrane (*Figure 1A–F*, purple density). Although in some tomograms two extracellular rings appeared to be present (see *Figure 1A and C*), only one ring was visible in the sub-tomogram average which could be either because one of the rings is more dynamic or substoichiometric

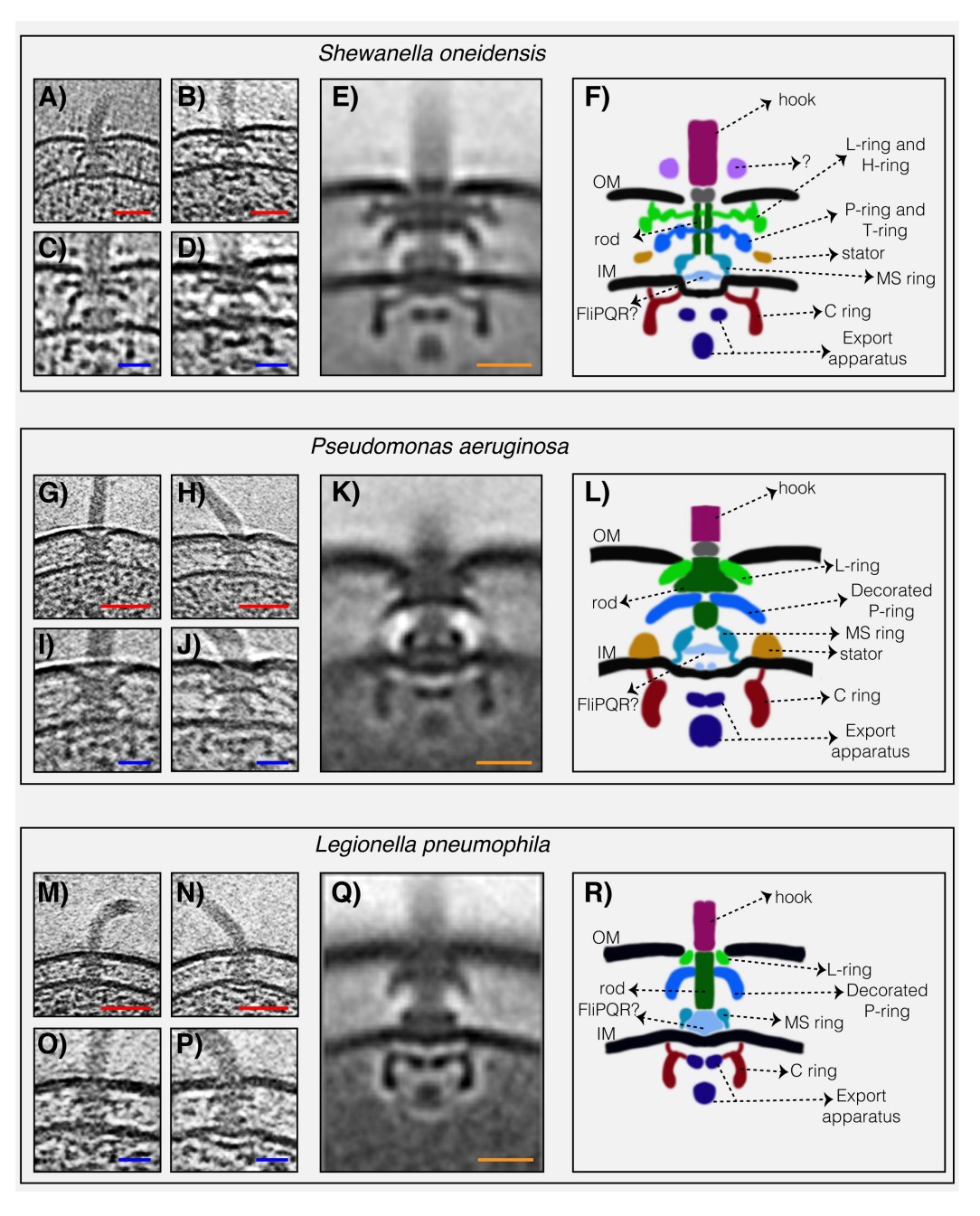

**Figure 1.** The structures of three dual-stator Gammaproteobacteria flagellar motors revealed by ECT. (**A and B**) slices through *Shewanella oneidensis* MR-1 electron cryo-tomograms showing single polar flagella. (**C and D**) zoomed-in views of the slices shown in (**A**) and (**B**) highlighting the flagellar motors. (**E**) central slice through a sub-tomogram average of the *S. oneidensis* MR-1 flagellar motor. (**F**) schematic representation of the sub-tomogram average shown in (**E**) with the major parts of the motor labeled. (**G–L**) flagellar motor of *Pseudomonas aeruginosa*. Panels follow the same scheme as in (**A–F**) above. (**M–R**) flagellar motor of *Legionella pneumophila*. Panels follow the same scheme as above. Scale bars 50 nm (red) and 20 nm (blue and orange).

DOI: https://doi.org/10.7554/eLife.43487.003

The following figure supplements are available for figure 1:

**Figure supplement 1.** Gold-standard FSC curves of sub-tomogram averages.

DOI: https://doi.org/10.7554/eLife.43487.004

**Figure supplement 2.** Slices through electron cryo-tomograms of *Shewanella oneidensis* MR-1 cells illustrating the presence of flagellar motors.

DOI: https://doi.org/10.7554/eLife.43487.005

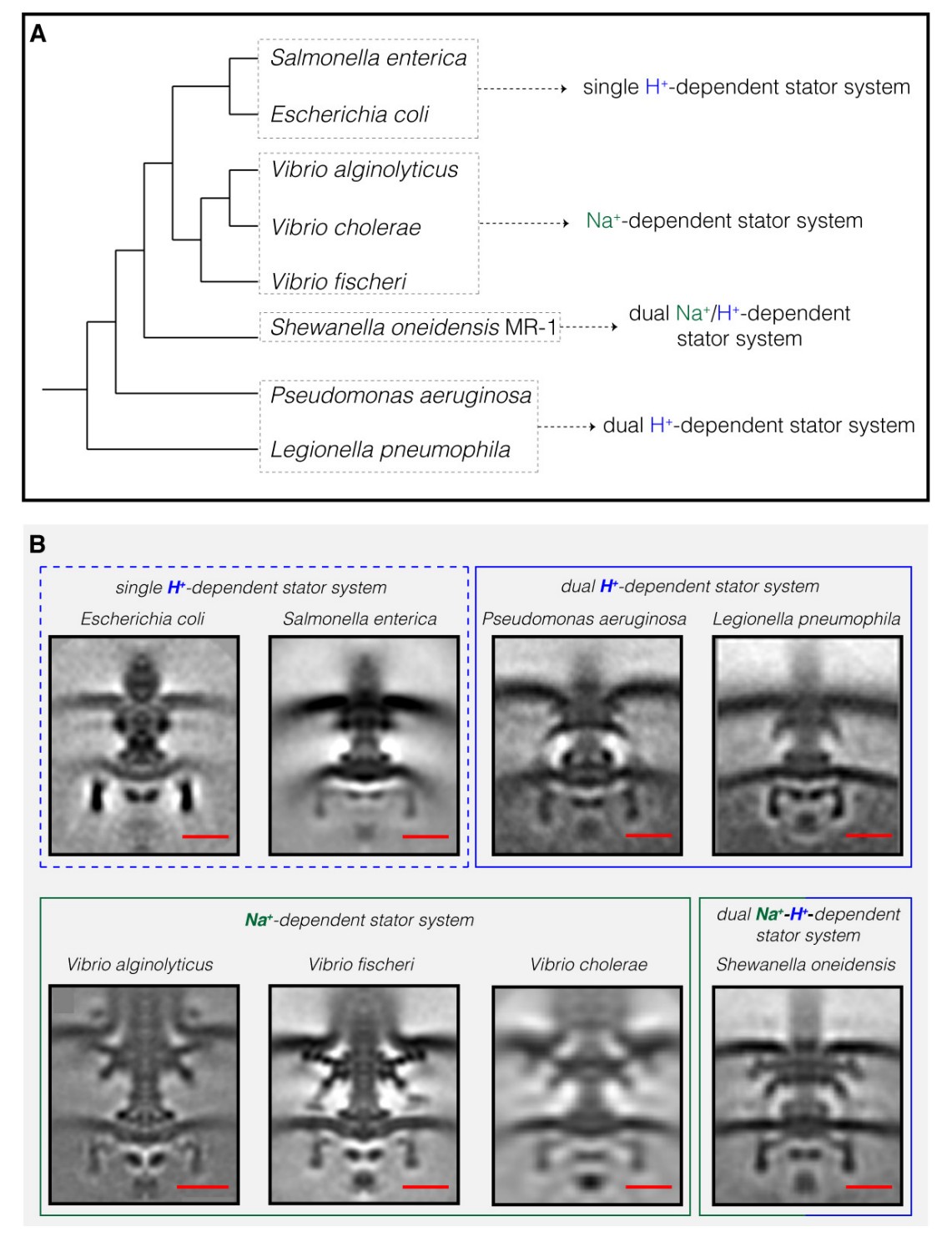

**Figure 2.** Compilation of all Gammaproteobacteria flagellar motors imaged to date by ECT. (**A**) A phylogenetic tree of the eight Gammaproteobacteria species with available ECT structures of their flagellar motors. This tree was made based on (*Williams et al., 2010*). (**B**) Central slices of sub-tomogram averages are shown for the eight Gammaproteobacteria flagellar motors revealed by ECT, including the three structures solved in this study (*P. aeruginosa*, *L. pneumophila* and *S. oneidensis*). The motors are classified based on their stator system: single $H^+$-driven (dashed blue box), dual $H^+$-driven (blue box), $Na^+$-driven (green box) or dual $Na^+$-$H^+$-driven (green-blue box). *E. coli* EMDB 5311, *S. enterica* EMDB 3154, *V. fischeri* EMDB 3155, *V. cholerae* EMDB 5308, *V. alginolyticus* is adapted from *Zhu et al., 2017*. Scale bars are 20 nm.
DOI: https://doi.org/10.7554/eLife.43487.006

**Table 1.** Candidate homologs of H- and T-ring components in species imaged in this study.

| Species | MotX candidate | MotY candidate | FlgO candidate | FlgP candidate | FlgT candidate |
|---|---|---|---|---|---|
| *Pseudomonas aeruginosa* (dual H$^+$-driven stator) | - | + 2e-37 (PA3526) | - | - | - |
| *Legionella pneumophila* (dual H$^+$-driven stator) | - | + 3e-35 (lpg2962) | - | - | - |
| *Shewanella oneidensis* MR-1 (dual Na$^+$-H$^+$-driven stator) | + 2e-46 (SO_3936) | + 2e-80 (SO_2754) | + 2e-19 (SO_3257) | + 6e-31 (SO_3256) | + 3e-36 (SO_3258) |

DOI: https://doi.org/10.7554/eLife.43487.007

(*Figure 1B,D* and *Figure 1—figure supplement 2* for more examples of single motors). This structure is reminiscent of the O-ring (outer membrane ring) described recently in the sheathed flagellum of *Vibrio alginolyticus* (*Zhu et al., 2017*). However, while the *V. alginolyticus* O-ring was associated with a 90° bend in the outer membrane, no such outer membrane bend was seen in the unsheathed *S. oneidensis* flagellum, so the function of this structure remains mysterious.

The most striking difference between the three motor structures was the L- and P-rings, which were highly elaborated in *S. oneidensis*. The *P. aeruginosa* and *L. pneumophila* motors lacked additional rings associated with the L-ring, but showed smaller elaborations of their P-rings. To determine whether flagellar motor structure correlates with stator type, we compared our three new ECT structures with those of the five previously-published Gammaproteobacteria motors (*Figure 2*). Two motors (*Escherichia coli* and *S. enterica*) have a single H$^+$-driven stator system, two motors have dual H$^+$-dependent stator systems (*P. aeruginosa* and *L. pneumophila*), three motors have Na$^+$-driven systems (the three *Vibrio* species) and one motor has a dual Na$^+$-H$^+$-driven system (*S. oneidensis*). Interestingly, we found that motors with similar stator type also shared similar structural characteristics. While the two motors with a single H$^+$-dependent stator system did not show any periplasmic elaborations beyond the conserved flagellar core, the dual H$^+$-dependent stator systems had an extra ring surrounding their P-ring, with no embellishment of the L-ring. The Na$^+$-dependent motors of the *Vibrio spp.*, together with the Na$^+$-H$^+$-dependent motor of *S. oneidensis,* have extra components surrounding both their P- and L- rings. In *Vibrio*, these extra periplasmic rings are known as the T-ring (surrounding the P- ring and formed by the MotX and MotY proteins) and the H-ring (surrounding the L-ring and consisting of the FlgO, FlgP and FlgT proteins). The presence of the T- and H-rings was suggested to be specific to the Na$^+$-driven *Vibrio* motors (*Minamino and Imada, 2015*) with the FlgT protein required for the formation of both rings (*Terashima et al., 2013*).

Previous studies showed that MotX and MotY are important for flagellar rotation in *S. oneidensis* but it was not known whether they form part of the motor or not (*Koerdt et al., 2009*). Similarly, bioinformatics analysis and biochemical studies showed that MotY is involved in the function of the *P. aeruginosa* motor, but the structural basis of this role was not known (*Doyle et al., 2004*). We

**Table 2.** Candidate homologs of H- and T-ring components in single H$^+$-dependent stator systems of Gammaproteobacteria.

| Species | MotX candidate | MotY candidate | FlgO candidate | FlgP candidate | FlgT candidate |
|---|---|---|---|---|---|
| *Escherichia coli* | - | - | - | - | - |
| *Salmonella enterica* | - | - | - | - | - |
| *Sodalis glossinidius* | - | - | - | - | - |
| *Photorhabdus laumondii subsp. laumondii TTO1* | - | - | - | - | - |
| *Serratia proteomaculans* | - | + 7e-13 (Spro_1787) | - | - | - |
| *Psychromonas ingrahamii* | - | + 3e-14 (Ping_3567) | - | - | - |

DOI: https://doi.org/10.7554/eLife.43487.008

**Table 3.** Candidate homologs of H- and T-ring components in dual H$^+$-dependent stator systems of Gammaproteobacteria.

| Species | MotX candidate | MotY candidate | FlgO candidate | FlgP candidate | FlgT candidate |
|---|---|---|---|---|---|
| *Azotobacter vinelandii* DJ | - | +<br>8e-14<br>(Avin_48650) | - | - | - |
| *Cellvibrio japonicas* Ueda107 | - | +<br>9e-28<br>(CJA_2588) | - | - | - |
| *Chromohalobacter salexigens* DSM 3043 | - | +<br>6e-13<br>(Csal_3309) | +<br>9e-16<br>(Csal_2511) | - | - |
| *Pseudomonas entomophila* | - | +<br>2e-31<br>(PSEEN1209) | - | - | - |
| *Saccharophagus degradans* 2–40 | - | +<br>1e-37<br>(Sde_2427) | - | - | - |
| *Xanthomonas campestris* pv. campestris | - | +<br>1e-13<br>(XCC1436) | - | - | - |
| *Pseudomonas putida* | - | +<br>7e-31<br>(PP_1087) | - | - | - |
| *Yersinia pestis* CO92 | - | +<br>6e-11<br>(YPO0448) | - | - | - |
| *Pseudomonas fluorescens* Pf0-1 | - | +<br>2e-30<br>(Pfl01_4518) | - | - | - |
| *Xanthomonas axonopodis* pv. citrumelo F1 | - | +<br>1e-14<br>(XACM_1468) | - | - | - |
| *Stenotrophomonas maltophilia* R551-3 | - | +<br>4e-10<br>(Smal_1563) | - | - | - |

DOI: https://doi.org/10.7554/eLife.43487.009

therefore performed a bioinformatics search for candidate homologs of MotX, MotY, FlgO, FlgP and FlgT in the genomes of *P. aeruginosa*, *L. pneumophila* and *S. oneidensis* to examine whether there is a correlation between the presence of homologous genes and the extra periplasmic rings observed in the ECT structures. While we found candidates for all five proteins constituting the T- and H-rings in *S. oneidensis* as previously suggested (*Wu et al., 2011*), only MotY candidates were found in *L. pneumophila* and *P. aeruginosa* (*Table 1*). This is in accordance with our ECT structures, which showed that *L. pneumophila* and *P. aeruginosa* motors have a ring surrounding only their P-rings while the *S. oneidensis* motor has rings surrounding both the P- and L-rings. These rings are likely T- and H-rings, respectively, as in *Vibrio*. The lack of candidate MotX homologs in the genomes of *L. pneumophila* and *P. aeruginosa* (*Table 1*) is consistent with their lack of PomB, the component of the Na$^+$-dependent stator with which MotX interacts. Interestingly, the absence of candidates for FlgT in the *L. pneumophila* and *P. aeruginosa* genomes suggests that it may not be required for the recruitment of MotY as in *Vibrio* species.

To see whether these correlations hold more broadly, we expanded our bioinformatics analysis to additional species of Gammaproteobacteria (*Williams et al., 2010*). We examined the genomes of species with single H$^+$-driven stator systems (*Table 2*), dual H$^+$-driven stator systems (*Table 3*) and Na$^+$-driven stator systems (*Table 4*). These species were identified either by Blasting the sequence of the stator proteins (MotA, B, C and D and Pom A and B, see Materials and methods) against the genome of the species or based on previous studies (*Thormann and Paulick, 2010*). In all species we examined, we observed the same pattern: (i) genomes of species with single H$^+$-driven stator

**Table 4.** Candidate homologs of H- and T-ring components in Na$^+$-dependent stator systems of Gammaproteobacteria.

| Species | MotX candidate | MotY candidate | FlgO candidate | FlgP candidate | FlgT candidate |
|---|---|---|---|---|---|
| Colwellia psychrerythraea 34H | +<br>2e-63<br>(CPS_4618) | +<br>1e-73<br>(CPS_3471) | +<br>2e-59<br>(CPS_1469) | +<br>6e-28<br>(CPS_1470) | +<br>5e-38<br>(CPS_1468) |
| Vibrio fischeri | +<br>1e-113<br>(VF_2317) | +<br>3e-141<br>(VF_0926) | +<br>3e-113<br>(VF_1884) | +<br>1e-60<br>(VF_1883) | +<br>2e-166<br>(VF_1885) |
| Vibrio vulnificus YJ016 | +<br>4e-136<br>(VV3065) | +<br>9e-177<br>(VV1183) | +<br>8e-140<br>(VV0953) | +<br>1e-77<br>(VV0954) | +<br>0.0<br>(VV0952) |
| Photobacterium profundum | +<br>1e-110<br>(PBPRA3344) | +<br>3e-146<br>(PBPRA2571) | +<br>5e-101<br>(PBPRA0894) | +<br>5e-60<br>(PBPRA0895) | +<br>6e-145<br>(PBPRA0893) |
| Pseudoalteromonas haloplanktis | +<br>3e-76<br>(PSHAa0276) | +<br>6e-73<br>(PSHAa2115) | +<br>3e-37<br>(PSHAa0755) | +<br>2e-26<br>(PSHAa0762) | +<br>5e-40<br>(PSHAa0761) |
| Pseudoalteromonas tunicata | +<br>4e-71<br>(PTUN_a0699) | +<br>1e-68<br>(PTUN_a1296) | +<br>2e-32<br>(PTUN_a3193) | +<br>2e-28<br>(PTUN_a3178) | +<br>4e-34<br>(PTUN_a3179) |
| Idiomarina loihiensis L2TR | +<br>5e-67<br>(IL2001) | +<br>4e-78<br>(IL1801) | +<br>4e-18<br>(IL1169) | +<br>9e-32<br>(IL1153) | +<br>1e-30<br>(IL1154) |
| Alteromonas macleodii ATCC 27126 | +<br>8e-73<br>(MASE_16945) | +<br>3e-74<br>(MASE_05600) | +<br>2e-34<br>(MASE_11745) | +<br>7e-29<br>(MASE_04615) | +<br>3e-35<br>(MASE_04610) |
| Pseudoalteromonas atlantica | +<br>2e-71<br>(Patl_0993) | +<br>1e-79<br>(Patl_1400) | +<br>1e-30<br>(Patl_1308) | +<br>1e-26<br>(Patl_3106) | +<br>1e-31<br>(Patl_3107) |

DOI: https://doi.org/10.7554/eLife.43487.010

systems lacked homologs of H- or T-ring components; (ii) genomes of species with Na$^+$ (or Na$^+$-H$^+$) stator systems contained homologs of all H- and T-ring components, and (iii) genomes of species with dual H$^+$-driven stator systems contained candidate homologs only for the T-ring component MotY. The sole exception to this rule was *Chromohalobacter salexigens* DSM 3043, which contained a homolog of FlgO in addition to MotY. Also, while *Serratia proteomaculans* and *Psychromonas ingrahamii* have candidates for single MotAB stator system they also have candidates for MotY (see *Supplementary file 1* and *2*). None of the thirteen species with dual H$^+$-driven stator systems we examined contained a homolog of FlgT, further suggesting that it is not essential for MotY stabilization in this group.

## Discussion

Together, our results from ECT imaging of flagellar motors in situ and bioinformatics analysis reveal a correlation between the structural elaboration of the flagellar motor of Gammaproteobacteria and the type of its torque-generating unit, the stator (summarized in *Figure 3*). Low-speed motors with single H$^+$-stator systems have only the P- and L-rings, while high-speed motors using Na$^+$ have two extra periplasmic rings, the T- and H-rings. Unexpectedly, we find that motors with dual H$^+$-driven stator systems represent a hybrid structure between the two, elaborating their P-rings with one of the five components of the T- and H-rings, MotY. It is important to note that the presence of these extra elaborations in the motor is encoded in the genome and is not related to whether or not a stator subunit is recruited on the motor. This extra MotY ring might help to stabilize the motor under conditions of increased load, as in the viscous environment of the pulmonary system encountered by *L. pneumophila* and *P. aeruginosa*. These results therefore suggest an evolutionary pathway in which these pathogenic Gammaproteobacteria species could have borrowed a motor stabilization strategy from related Na$^+$-driven motors to allow them to colonize animal hosts. Finally, It would be interesting to investigate whether our observation here holds for other bacterial species that use different

**Figure 3.** Models showing correlation between structural elaboration of the flagellar motor and its stator type. Flagellar motors with single H$^+$-driven stator systems (e.g. *Salmonella*) have P- and L-rings alone. Motors with dual H$^+$-driven stator systems have an extra ring surrounding the P-ring formed by the MotY protein alone. Motors with Na$^+$-driven motors have two periplasmic rings, the T-ring (MotX and MotY) and H-ring (FlgO, FlgP and FlgT), decorating the P- and L-rings, respectively. Note that the boundaries between the P- and L-rings and their decorations are tentative in these schematics.

DOI: https://doi.org/10.7554/eLife.43487.011

cations as their energy source (*Ito and Takahashi, 2017*) and whether it extends to other bacterial species with more than two stator systems or other classes.

# Materials and methods

## Strains and growth conditions

*Legionella pneumophila* (strain Lp02) cells were grown on plates of ACES [N-(2-acetamido)−2-aminoethanesulfonic acid]-buffered charcoal yeast extract agar (CYE) or in ACES-buffered yeast extract broth (AYE) with 100 µg/ml thymidine. Ferric nitrate and cysteine hydrochloride were added to the media. For ECT experiments, cells were harvested in early stationary phase.

*Shewanella oneidensis* MR-1 cells belonging to the strains listed in *Supplementary file 3* were used in this study. They were grown using one of the following methods: Luria–Bertani (LB) broth culture, chemostat, the batch culture method or in a perfusion flow imaging platform. Detailed descriptions of these methods can be found in *Subramanian et al. (2018)*. Briefly, in the chemostat method, 5 mL of a stationary-phase overnight LB culture was injected into a continuous flow bioreactor containing an operating liquid volume of 1 L of a defined medium (*Pirbadian et al., 2014*), while dissolved oxygen tension (DOT) was maintained at 20%. After 20 hr, and as the culture reached stationary phase, continuous flow of the defined medium (*Pirbadian et al., 2014*) was started with a dilution rate of 0.05 hr$^{-1}$ while DOT was still maintained at 20%. After 48 hr of aerobic growth under continuous flow conditions, the DOT was manually reduced to 0%. O$_2$ served as the sole terminal electron acceptor throughout the experiment. pH was maintained at 7.0, temperature at 30°C, and agitation at 200 rpm. Either 24 or 40 hr after DOT reached 0%, samples were taken from the chemostat for ECT imaging.

In the batch culture method, 200 µL of an overnight LB culture of *S. oneidensis* cells was added to each of two sealed and autoclaved serum bottles containing 60 mL of a defined medium (*Pirbadian et al., 2014*). One of the two bottles acted as a control and was not used for imaging. To this control bottle, 5 µM resazurin was added to indicate the O$_2$ levels in the medium. The bottles were then placed in an incubator at 30°C, with shaking at 150 rpm until the color due to resazurin in the control bottle completely faded, indicating anaerobic conditions. At this point, samples were taken for ECT imaging from the bottle that did not contain resazurin.

For the perfusion flow imaging experiments, *S. oneidensis* cells were grown overnight in LB broth at 30°C to an OD$_{600}$ of 2.4–2.8 and washed twice in a defined medium (*Pirbadian et al., 2014*). A glow-discharged, carbon-coated, R2/2, Au NH2 London finder Quantifoil EM grid was glued to a 43 mm × 50 mm no. 1 glass coverslip using waterproof silicone glue (General Electric Company) and let dry for ∼ 30 min. Using a vacuum line, the perfusion chamber (model VC-LFR-25; C and L

Instruments) was sealed against the grid-attached glass coverslip. A total of ~10 mL of the washed culture was injected into the chamber slowly to allow cells to settle on the grid surface, followed by a flow of sterile defined medium from an inverted serum bottle through a bubble trap (model 006BT-HF; Omnifit) into the perfusion chamber inlet. Subsequently, the flow of medium was stopped and the perfusion chamber was opened under sterile medium. The grid was then detached from the coverslip by scraping off the silicone glue at the grid edges using a 22-gauge needle and rinsed by transferring three times in deionized water, before imaging by ECT.

Samples were also prepared from an aerobic *S. oneidensis* LB culture grown at 30°C to an $OD_{600}$ of 2.4–2.8.

*Pseudomonas aeruginosa* PAO1 cells were first grown on LB plates at 37°C overnight. Subsequently, cells were inoculated into 5 ml MOPS [(3-(*N*-morpholino) propanesulfonic acid)] Minimal Media Limited Nitrogen and grown for ~24 hr at 30°C.

Many of the flagellar motors analyzed here were taken from tomograms recorded for more than one purpose. The *Shewanella oneidensis* MR-1 mutants, for instance, were grown under different growth conditions for the purpose of studying the nanowires formed by these cells (see *Subramanian et al., 2018*), but all their motors were presumably the same, so we included them here to increase the clarity and resolution of our average.

## Sample preparation for electron cryo-tomography

Cells (*L. pneumophila*, *P. aeruginosa* and *S. oneidensis*) from batch cultures and chemostats were mixed with BSA (Bovine Serum Albumin)-treated 10 nm colloidal gold solution (Sigma-Aldrich, St. Louis, MO, USA) and 4 µL of this mixture was applied to a glow-discharged, carbon-coated, R2/2, 200 mesh copper Quantifoil grid (Quantifoil Micro Tools) in a Vitrobot Mark IV chamber (FEI). Excess liquid was blotted off and the grid was plunge frozen in a liquid ethane/propane mixture for ECT imaging.

## Electron cryo-tomography

Imaging of ECT samples (*S. oneidensis* and *P. aeruginosa*) was performed on an FEI Polara 300-keV field emission gun electron microscope (FEI company, Hillsboro, OR, USA) equipped with a Gatan image filter and K2 Summit counting electron-detector camera (Gatan, Pleasanton, CA, USA). Data were collected using the UCSF Tomography software (*Zheng et al., 2007*), with each tilt series ranging from −60° to 60° in 1° increments, an underfocus of ~ 5–10 µm, and a cumulative electron dose of ~ 130–160 $e^-/A^2$ for each individual tilt series. For *L. pneumophila* samples, imaging was done using an FEI Titan Krios 300 kV field emission gun transmission electron microscope equipped with a Gatan imaging filter and a K2 Summit direct electron detector in counting mode (Gatan). *L. pneumophila* data was also collected using UCSF Tomography software and a total dose of ~ 100 $e^-/A^2$ per tilt series with ~ 6 um underfocus.

## Sub-tomogram averaging

The IMOD software package was used to calculate three-dimensional reconstructions of tilt series (*Kremer et al., 1996*). Alternatively, the images were aligned and contrast transfer function corrected using the IMOD software package before producing SIRT reconstructions using the TOMO3D program (*Agulleiro and Fernandez, 2011*). Sub-tomogram averages with 2-fold symmetrization along the particle Y-axis were produced using the PEET program (*Nicastro et al., 2006*). To obtain the sub-tomogram averages of the flagellar motors we reconstructed 156 tomograms of *Pseudomonas aeruginosa*, 50 of *Legionella pneumophila* and ~ 300 of *Shewanella oneidensis* MR-1. The averages were obtained by averaging 144 sub-volumes *P. aeruginosa*, 100 sub-volumes *S. oneidensis* MR-1 and 45 sub-volumes *L. pneumophila*.

## Bioinformatics analysis

Candidate H- and T-ring component genes were identified by sequence alignment of the following *Vibrio cholerae* proteins against the fully sequenced genomes of each bacterial species using BLASTP (https://www.genome.jp/tools/blast/). The *Vibrio cholerae* proteins used were: MotX (Q9KNX9), MotY (Q9KT95), FlgO (Q9KQ00), FlgP (Q9KQ01) and FlgT (Q9KPZ9). To check for the stator system candidates in different species, the following proteins were blasted against the

genome of the bacterial species: PomAB proteins of *V. cholerae* (Q9KTL0 and Q9KTK9 respectivley), MotAB proteins of *E. coli* (P09348 and P0AF06 respectively) and MotCD of *P. aeruginosa* (G3XD73 and G3XD90 respectively) using BLASTP. Candidate MotX and MotY homologs identified were adjacent to the flagellar cluster in the genome, and for each stator system candidate homologs were characteristically located in tandem in the genome. The codes in parentheses represent Uniprot IDs. An *E*-value cutoff of $<1\times10^{-10}$ was used. The raw BLAST results for all species are shown in *Supplementary file 1* and *2*. Note that for the stator system, a candidate stator locus was considered only when two neighboring candidates for Mot/B, MotC/D or PomA/B were found.

## Acknowledgements

We thank Dr. Songye Chen for technical support. This work is supported by the National Institutes of Health (NIH, grant R01 AI127401 to GJJ). MK is supported by a postdoctoral Rubicon fellowship from De Nederlandse Organisatie voor Wetenschappelijk Onderzoek (NWO). SP and MYE-N. are supported by the Air Force Office of Scientific Research Presidential Early Career Award for Scientists and Engineers (FA955014-1-0294, to MYE-N). ECT was performed in the Beckman Institute Resource Center for Cryo-EM.

# Additional information

### Funding

| Funder | Grant reference number | Author |
| --- | --- | --- |
| National Institutes of Health | R01 AI127401 | Mohammed Kaplan<br>Debnath Ghosal<br>Poorna Subramanian<br>Catherine M Oikonomou<br>Andreas Kjaer<br>Davi R Ortega<br>Ariane Briegel<br>Grant J Jensen |
| Nederlandse Organisatie voor Wetenschappelijk Onderzoek | Rubicon fellowship | Mohammed Kaplan |
| Air Force Office of Scientific Research | FA955014-1-0294 | Sahand Pirbadian<br>Mohamed Y El-Naggar |

The funders had no role in study design, data collection and interpretation, or the decision to submit the work for publication.

### Author contributions

Mohammed Kaplan, Conceptualization, Data curation, Formal analysis, Funding acquisition, Validation, Investigation, Visualization, Methodology, Writing—original draft, Writing—review and editing; Debnath Ghosal, Poorna Subramanian, Ariane Briegel, Data curation, Writing—review and editing; Catherine M Oikonomou, Andreas Kjaer, Formal analysis, Writing—review and editing; Sahand Pirbadian, Davi R Ortega, Methodology, Writing—review and editing; Mohamed Y El-Naggar, Resources, Writing—review and editing; Grant J Jensen, Conceptualization, Resources, Formal analysis, Supervision, Funding acquisition, Validation, Investigation, Project administration, Writing—review and editing

### Author ORCIDs

Mohammed Kaplan (ID) http://orcid.org/0000-0002-0759-0459
Andreas Kjaer (ID) https://orcid.org/0000-0002-0096-5764
Ariane Briegel (ID) https://orcid.org/0000-0003-3733-3725
Grant J Jensen (ID) http://orcid.org/0000-0003-1556-4864

### Decision letter and Author response

Decision letter https://doi.org/10.7554/eLife.43487.023

Author response https://doi.org/10.7554/eLife.43487.024

## Additional files

### Supplementary files

• Supplementary file 1. Raw T- and H-rings proteins Blast results for all species in *Tables 1–4*.
DOI: https://doi.org/10.7554/eLife.43487.012

• Supplementary file 2. Raw stator proteins Blast results for all species in *Tables 1–4*.
DOI: https://doi.org/10.7554/eLife.43487.013

• Supplementary file 3. *S. oneidensis* strains used in this study.
DOI: https://doi.org/10.7554/eLife.43487.014

• Transparent reporting form
DOI: https://doi.org/10.7554/eLife.43487.015

### Data availability

All data generated or analysed during this study are included in the manuscript and supporting files. The ECT structures have been deposited in the EMDB under the following accession numbers, EMD-0464 for *Legionella pneumophila* motor, EMD-0465 for *Pseudomonas aeruginosa* motor and EMD-0467 for *Shewanella oneidensis* MR-1 motor.

The following datasets were generated:

| Author(s) | Year | Dataset title | Dataset URL | Database and Identifier |
|---|---|---|---|---|
| Mohammed Kaplan, Debnath Ghosal, Poorna Subramanian, Catherine M Oikonomou, Andreas Kjaer, Sahand Pirbadian, Davi R Ortega, Ariane Briegel, Mohamed Y El-Naggar, Grant J Jensen | 2019 | *Legionella pneumophila* motor | http://www.ebi.ac.uk/pdbe/entry/emdb/EMD-0464 | EM Data Bank, EMD-0464 |
| Mohammed Kaplan, Debnath Ghosal, Poorna Subramanian, Catherine M Oikonomou, Andreas Kjaer, Sahand Pirbadian, Davi R Ortega, Ariane Briegel, Mohamed Y El-Naggar, Grant J Jensen | 2019 | *Shewanella oneidensis* MR-1 motor | http://www.ebi.ac.uk/pdbe/entry/emdb/EMD-0467 | EM Data Bank, EMD-0467 |
| Mohammed Kaplan, Debnath Ghosal, Poorna Subramanian, Catherine M Oikonomou, Andreas Kjaer, Sahand Pirbadian, Davi R Ortega, Ariane Briegel, Mohamed Y El-Naggar, Grant J Jensen | 2019 | *Pseudomonas aeruginosa* motor | http://www.ebi.ac.uk/pdbe/entry/emdb/EMD-0465 | EM Data Bank, EMD-0465 |

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
