## [Decision Letter]

[Editors’ note: a previous version of this study was rejected after peer review, but the authors submitted for reconsideration, and the paper was reviewed and accepted. The original decision letter after peer review is shown below.]

Thank you for submitting your work entitled "Structural complexity of the Gammaproteobacteria flagellar motor is related to the type of its torque-generating stators" for consideration by *eLife*. Your article has been reviewed by three peer reviewers, one of whom is a member of our Board of Reviewing Editors, and the evaluation has been overseen by a Senior Editor. The following individuals involved in review of your submission have agreed to reveal their identity: Masahiro Ito (Reviewer #2); Henning Stahlberg (Reviewer #3).

Our decision has been reached after consultation between the reviewers. Based on these discussions and the individual reviews below, we regret to inform you that your work will not be considered further for publication in *eLife*. Although the present manuscript is being rejected, the reviewers felt that a substantially improved paper might certainly be considered again for publication in *eLife*.

*Reviewer #1:*

While some of the findings appear interesting, my main concern was that the resolution is too low to be able to reach any mechanistic conclusions, and seeing the difference between one ring and two as due to "increased complexity" may not make any sense. The structure with one ring might actually be the more complex one. It would be like saying that a ribosome that appears as two blobs is more complex than one with one blob. Are viruses with two (or five) different capsid proteins more complex than those with only one? In some cases it might be shown that evolution has led to a reduction in the number of different subunits. I thus had an overall limited degree of enthusiasm for this paper.

*Reviewer #2:*

This paper observed with electron cryo-tomography (ECT) the bacterial flagellar motor of three kinds of Γ-Proteobacteria and compared with the ECT image of each flagella motor which has been published so far, and then grouping was carried out. At the same time, the authors attempted to clarify whether there is a correlation between the bound ion of the flagellar stator and the ECT image by incorporating the bioinformatics approach.

First, this reviewer disagrees with the title. Bacterial flagellar motors have been reported to alter the coupling ions used depending on the growth environment (For example, Terahara, Krulwich and Ito, 2008, and Terahara, Sano and Ito, 2012) Because there is no experimental direct evidence with just the linkage by the flagellar motor of the Γ-Proteobacteria (and there is one exception among the eight surveyed cases), the authors should give a title to the extent that they can assert from the experimental results. Also, although there is a case where bioinformatics was utilized, the main manuscript feels uncomfortable for me because the data are not presented in all supplemental data. Furthermore, in order to satisfy the reader, it is necessary to use a phylogenetic tree to draw a diagram showing the relationship between each strain and its motor type.

(H^+^ or Na+) – this reviewer disagrees with this expression. Currently, motors driven by K^+^ and divalent cations are also reported (For example, Ito and Takahashi, 2017) The authors should mention and cite them as well.

Is there any example of enzymes and channels that can be classified by structure and coupling ions?

In this paper, do the authors think that the structure of the flagellar motor is forthcoming or whether the type of stator prescribes the structure? Do the authors discuss it in the text?

Figure 1The flagellar structure of Shewanella and *Pseudomonas* can confirm the stator. However, in Legionella pneumophila, the stator is invisible. Does one need to think that it affects the structure of the flagellar motor with or without a stator?

Table 1, 2, 3: As for the symbol "+", the authors should state the access number of each protein in the table. If the access number is not stated, the reviewer cannot evaluate whether or not it is correct.

Table 1, 2, 3: At the end of each title …dependent stations systems of gammaproteobacteria – it should be attached accurately.

Supplementary file 3: The authors do not explain where these mutants were used in the text. Please explain properly. Also, this reviewer does not know why the authors only described *S. oneidensis* and its derivative mutants.

*Reviewer #3:*

Kaplan et al. describe a structural analysis of the flagellar motor systems of three Gammaproteobacteria that all have so-called dual-stator motors: These three species all have flagellar motors that can be driven by both, either by H^+^ gradients or by Na^+^ gradients.

Stators are protein components coupled to the inner membrane of the bacteria, while the rotating parts of the flagellar motors are components formed by other protein parts. Flagellar systems are rotating the rotor with respect to the stator with the help of motor proteins.

While some (more simple) flagellar systems have stators that can only be driven by H^+^, others have stators that can only be driven by Na+, while more advanced flagellar motors have dual stators so that they can be driven by both types of fuel. These bacteria make use of their "flex-fluel motors", when they need to swim in environments with strongly different viscosities.

The authors have used cryo-ET (ECT) to determine the structures of three motors at low resolution (~6 nm) by sub-volume averaging. They have also performed bioinformatics analysis of gene homologies to find candidate proteins for different motor components in different bacterial genomes. They find that bacteria with similar stator types haver similar structural motifs. They also find that bacteria that have flagellar systems with dual stators (H^+^ and Na^+^ on the same time) also have a higher complexity in their motor component constructions.

The manuscript is rather short and compact, but well written and interesting to read.

[Minor comments not shown.]

---

## [Author Response]

In response to the reviewers’ comments, we have revised the text, the figures and performed new bioinformatics analysis to include more than double the number of species analyzed in our first manuscript. We feel that we have adequately addressed all the reviewers’ concerns and request reconsideration.

Reviewer #1:

While some of the findings appear interesting, my main concern was that the resolution is too low to be able to reach any mechanistic conclusions, and seeing the difference between one ring and two as due to "increased complexity" may not make any sense. The structure with one ring might actually be the more complex one. It would be like saying that a ribosome that appears as two blobs is more complex than one with one blob. Are viruses with two (or five) different capsid proteins more complex than those with only one? In some cases it might be shown that evolution has led to a reduction in the number of different subunits. I thus had an overall limited degree of enthusiasm for this paper.

We agree – this is a good point – the presence of additional proteins is not always the same as increased complexity. We have now changed the title and removed all other statements about “complexity.” The new title is “The presence and absence of periplasmic rings in bacterial flagellar motors correlates with stator type.”

Reviewer #2:

[…] First, this reviewer disagrees with the title. Bacterial flagellar motors have been reported to alter the coupling ions used depending on the growth environment (For example, Terahara, Krulwich and Ito, 2008, and Terahara, Sano and Ito, 2012) Because there is no experimental direct evidence with just the linkage by the flagellar motor of the Γ-Proteobacteria (and there is one exception among the eight surveyed cases), the authors should give a title to the extent that they can assert from the experimental results.

We have now changed the title of the paper (see above), and have added to the text:

“The bacterial flagellum is a macromolecular machine that transforms the movement of ions (H^+^, Na^+^ or other cations) across the cell membrane into a mechanical torque to move the bacterial cell through its environment (Ito and Takahashi, 2017; Sowa and Berry, 2008).”

We also added the following to the Discussion part:

“Finally, It would be interesting to investigate whether our observation here holds for other bacterial species that uses different cations as their energy source (Ito and Takahashi, 2017) and whether it extends to other bacterial species with more than two stator systems or other classes”

Also, although there is a case where bioinformatics was utilized, the main manuscript feels uncomfortable for me because the data are not presented in all supplemental data.

We think the reviewer’s concern is not in the arguments or data presented, but that the tables were not shown in the main paper (they were only shown in the supplementary materials). We have now extended our bioinformatics analysis to include another 16 bacterial species which also supports our original claims. Also, the data is now included in the main text as Tables (1-4), while the raw Blast results are shown in Supplementary file 1 and 2.

Furthermore, in order to satisfy the reader, it is necessary to use a phylogenetic tree to draw a diagram showing the relationship between each strain and its motor type.

A tree is now presented as Figure 2A, in the main text.

(H^+^ or Na+) – this reviewer disagrees with this expression. Currently, motors driven by K^+^ and divalent cations are also reported (For example, Ito and Takahashi, 2017) The authors should mention and cite them as well.

We thank the reviewer for this important point. We have modified the text and added references as follows:

In the Introduction:

1) “The bacterial flagellum is a macromolecular machine that transforms the movement of ions (H^+^, Na^+^ or other cations) across the cell membrane into a mechanical torque to move the bacterial cell through its environment (Ito and Takahashi, 2017; Sowa and Berry, 2008).”

2) “Interestingly, it has also been shown that the single stator system of *Bacillus clausii* KSM-K16 is able to use both Na^+^ and H^+^ at different pH levels (Terahara et al., 2008). Additionally, some species (like alkaliphilic *Bacillus alcalophilus* AV1934 and *Paenibacillus* sp. TCA20), can use other cations to generate the energy required for torque generation depending on their environment (Ito and Takahashi, 2017; Terahara et al., 2012).”

We also modified the Discussion part (see above, first point of second reviewer).

Is there any example of enzymes and channels that can be classified by structure and coupling ions?In this paper, do the authors think that the structure of the flagellar motor is forthcoming or whether the type of stator prescribes the structure? Do the authors discuss it in the text?

Although ion channels share common features (they are all integral membrane proteins, for instance), their structures vary considerably, for sure in part because of differences in the ions they transmit. We don’t think that this is directly comparable to our observations here, however, because we are reporting correlations between stator types and simply the presence or absence of separate support rings (completely different proteins) in a very large macromolecular machine. Concerning the latter questions, we think the reviewer is interested in cause and effect: do different stators require different support rings, or do different support rings govern which stator is used? Our data do not answer this question, so we don’t discuss it in the text, but our suspicion is that because different environments have different ions available, and because different biological niches require higher- or lower-torque motors, the environment and niche select for certain stator/motor combinations. We guess that the higher torque stators provide a greater fitness advantage when they are stabilized by more elaborate support rings.

Figure 1: The flagellar structure of Shewanella and Pseudomonas can confirm the stator. However, in Legionella pneumophila, the stator is invisible. Does one need to think that it affects the structure of the flagellar motor with or without a stator?

The stator is a dynamic part of the motor. In other words, individual stators move in and out of the motor (see e.g., (Chen et al., 2011; Zhu et al., 2017)). The data reveal that stators are often found above the C-rings in *Shewanella* and *Pseudomonas* (and are therefore visible in the subtomogram averages), but are either often not present or not consistently positioned in *Legionella* (and are therefore not seen in the averages). But the presence or absence of H- and T-rings is encoded in the genome, as governed by natural selection.

We have now clarified all this in the revised text.

“While the *S. oneidensis* and *P. aeruginosa* averagesshowed clear densities corresponding to the stators (Figure 1E, F, K and L, orange density), none were visible in the *L. pneumophila* average, suggesting that they were more variable, or dynamic and therefore are not visible in the average (see e.g., (Chen et al., 2011; Zhu et al., 2017))”.

Also, we added the following to the Discussion part:

“It is important to note that the presence of these extra elaborations in the motor is encoded in the genome and is not related to whether or not a stator subunit is recruited on the motor.”

Table 1, 2, 3: As for the symbol "+", the authors should state the access number of each protein in the table. If the access number is not stated, the reviewer cannot evaluate whether or not it is correct.

The accession numbers of the proteins are now stated with the main Tables (1-4) as well as the raw data in Supplementary file 1 and 2.

Table 1, 2, 3: At the end of each title …dependent stations systems of gammaproteobacteria – it should be attached accurately.

Done.

Supplementary file 3: The authors do not explain where these mutants were used in the text. Please explain properly. Also, this reviewer does not know why the authors only described S. oneidensis and its derivative mutants.

Thank you for calling our attention to this admittedly confusing point. We have now tried to clarify this in the Materials and methods section as follows:

“Many of the flagellar motors analyzed here were taken from tomograms recorded for more than one purpose. The *Shewanella oneidensis* MR-1 mutants, for instance, were grown under different growth conditions for the purpose of studying the nanowires formed by these cells (see Subramanian et al., 2018), but all their motors were presumably the same, so we included them here to increase the clarity and resolution of our average.”